# Effect of Supplemental *Kluyveromyces marxianus* and *Pichia kudriavzevii* on Aflatoxin M_1_ Excretion in Milk of Lactating Dairy Cows

**DOI:** 10.3390/ani10040709

**Published:** 2020-04-18

**Authors:** Malinee Intanoo, Mallika B. Kongkeitkajorn, Witaya Suriyasathaporn, Yupin Phasuk, John K. Bernard, Virote Pattarajinda

**Affiliations:** 1Department of Animal Science, Faculty of Agriculture, Khon Kaen University, Khon Kaen 40002, Thailand; malineekku@hotmail.com (M.I.); yuplua@kku.ac.th (Y.P.); 2Department of Biotechnology, Faculty of Technology, Khon Kaen University, Khon Kaen 40002, Thailand; mallikab@kku.ac.th; 3Fermentation Research Center for Value Added Agricultural Products, Faculty of Technology, Khon Kaen University, Khon Kaen 40002, Thailand; 4Department of Feed Animal Clinics, Faculty of Veterinary Medicine, Chiang Mai University, Chiang Mai 50100, Thailand; suriyasathaporn@hotmail.com; 5Department of Animal and Dairy Science, College of Agricultural and Environmental Science, The University of Georgia, Tifton, GA 31793, USA; jbernard@uga.edu

**Keywords:** aflatoxins, mycotoxins, yeasts, feed supplement, dairy cows

## Abstract

**Simple Summary:**

A recent survey determining the occurrence of mycotoxins showed that almost all feedstuffs fed to dairy cattle contained aflatoxin, predominantly B_1_ type. The present study illustrated the potential application of aflatoxin-detoxifying yeast isolated from ruminal fluid of dairy cows to enhance the aflatoxin B_1_ detoxification in the rumen, to reduce the aflatoxin M_1_ contamination in milk and to improve dairy cattle performances. The inclusion of 2 g/day yeast into total mixed ration (TMR) diet reduced the transfer of aflatoxin B_1_ to aflatoxin M_1_ in raw milk by 72.08% and negative effects of aflatoxin B_1_ on dry matter intake (DMI) and milk compositions. Aflatoxin-detoxifying yeast isolates could potentially be developed for use as a feed additive to reduce aflatoxin contamination in milk and dairy products.

**Abstract:**

The objective of this study was to determine the effect of supplementing *Kluyveromyces marxianus* CPY1, *K. marxianus* RSY5 and *Pichia kudriavzevii* YSY2 isolated from ruminal fluid of dairy cows on transfer of aflatoxin B_1_ (AFB_1_) from feed into aflatoxin M_1_ (AFM_1_) in milk, DMI, milk production and nutrient digestibility. Four multiparous Holsteins in mid-lactation were used in a 4 × 4 Latin square design trial consisting of 14 days in each experimental period for sample collection. Between each period, 14 clearance days prior to the next treatment were allowed to minimize carryover effects. In each treatment, subsequent supplementation of isolated yeast was compared, i.e., (1) control (without yeast supplementation), (2) *K. marxianus* CPY1 (K1Y), (3) *K. marxianus* RSY5 (K2Y) and (4) *P. kudriavzevii* YSY2 (PY). All diets contained 22.28 µg of AFB_1_/kg. Treatments were individually fed at the rate of 2 g/day (1 × 10^9^ CFU/g) of yeast biomass or corn meal in the control group. Concentrations of AFM_1_ in milk was reduced with yeast and averaged 1.54, 0.36, 0.43 and 0.51 µg/L for control, K1Y, K2Y and PY, respectively (*p* < 0.01). The transfer of AFB_1_ from feed into AFM_1_ in milk was higher in control compared with K1Y, K2Y and PY (7.26% vs. 1.18%, 1.44% and 1.69% respectively, *p* < 0.01). Supplementation of yeast also improved DMI and milk compositions, but no differences were observed in nutrient digestibility or milk yield among treatments. Concentration and yield of milk protein, fat, lactose, solid-not-fat (SNF) and total solids were greater in cows fed yeast compared with the control (*p* < 0.01). These results indicate that *K. marxianus* CPY1, RSY5 and *P. kudriavzevii* YSY2 shows promise as a dietary supplementation to detoxify AFB_1_ and improve DMI and yield of milk components.

## 1. Introduction

There has been an upward trend in the use of agro-industrial byproducts as a source of nutrients for feeding dairy cattle. However, most feedstuffs are prone to being contaminated with mycotoxins, especially aflatoxins, which are a secondary metabolite produced mainly by *Aspergillus flavus* and *A. parasiticus*. The family of compounds includes aflatoxin B_1_, B_2_, G_1_ and G_2_ (AFB_1_, AFB_2_, AFG_1_ and AFG_2_) that can directly contaminate food and animal feed. Two additional metabolic products, aflatoxin M_1_ and M_2_ (AFM_1_ and AFM_2_), are often found in milk from cows consuming feeds contaminated with aflatoxin. The AFB_1_ is the most acutely toxic type of aflatoxin compound and is a member of group 1 carcinogenic mycotoxins which pose serious problems in human health and negatively impact agricultural economics [1]. The AFM_1_ is the 4-hydroxy derivative of AFB_1_, formed in the liver and excreted in the milk by the mammary glands of both humans and lactating animals fed diets containing AFB_1_-contaminated ingredients [2]. In dairy cattle, ingested AFB_1_ is biotransformed in the liver to AFM_1_ which is then distributed throughout tissues, milk and biological fluids of the animal. In addition, when dairy cattle consume diets contaminated with 200 or more µg/kg of AFB_1_ for a long period of time, reductions in feed intake, growth rate, lactation and vaccine-induced immunity are observed [3,4,5].

Feeding dairy cattle AFB_1_-contaminated diets results in contamination of milk with AFM_1_ within 12 to 24 h [3]. The presence of AFM_1_ in raw milk and dairy products is an important food safety issue because of their high consumption by humans, especially children [6]. Regulatory authorities have set limits on allowable milk AFM_1_ concentration, particularly for infants, who consume appreciable amounts of milk [6,7]. For example, the U.S. Food and Drug Administration (FDA) has established 0.50 µg/kg as the maximum permissible amount of AFM_1_ in milk. The corresponding AFB_1_ limit in diets fed to lactating dairy cows is 20 µg/kg. In contrast, the maximum limit of AFM_1_ in milk allowed by the European Commission is 0.05 µg/kg [8]. Thailand has adopted the U.S. FDA limits for AFB_1_ in feed and AFM_1_ in milk.

Several studies have reported the effectiveness of mycotoxin degradation in feeds using physical, chemical or biological control methods [9,10,11,12,13,14]. For biological control, several bacteria species such as *Bacillus*, *Pseudomonas* and *Lactobacilli* spp. have been reported to inhibit growth of aflatoxin-producing *Aspergillus* spp. [15,16,17,18,19,20]. Yeast has also been used in some research to help animal performance and ruminal metabolism and to detoxify mycotoxins in feed [16,21]. Results of these studies have demonstrated that *Saccharomyces cerevisiae*, *Pichia anomala* and *Candida krusei* can reduce mycotoxin contamination in feed and hence minimize the health risk of livestock consuming contaminated feed ingredients. Yeast is a naturally occurring microorganism frequently used in the food and beverage industries [2]. Additionally, due to yeast’s ability to bind and enzymatically degrade mycotoxins [22,23,24,25], biotransformation and detoxification of mycotoxins by yeast or its enzymes can serve as a method to effectively and safely control mycotoxins [25,26].

In our previous study, we isolated AFB_1_-detoxifying yeast from ruminal fluid of lactating dairy cows and used as a probiotic and mycotoxin biodegradation source. The results revealed that yeast isolates *K. marxianus* CPY1, *K. marxianus* RSY5 and *P. kudriavzevii* YSY2 can effectively detoxify AFB_1_ in vitro [27]. The objectives of this study were to determine the effects of supplementing *K. marxianus* CPY1, *K. marxianus* RSY5 and *P. kudriavzevii* YSY2 on transferring of AFB_1_ in feed into AFM_1_ in cow’s milk and on dry matter intake (DMI), nutrient digestibility and milk production of lactating dairy cows fed these supplements.

## 2. Materials and Methods

The protocols used in this study were approved by the Animal Ethics Committee of Khon Kaen University (approval no. ACUC-KKU-59/2560), based on the Ethic of Animal Experimentation of National Research Council of Thailand.

### 2.1. Experimental Procedures

This experiment was conducted at the Roi Et Agricultural research and training center located in Roi Et province, Thailand, from June to September 2017. The herd consisted of 213 lactating cows with an average milk yield of 12 kg/day. Four multiparous Holstein cows in mid-lactation (lactation number = 3, 180 ± 21 days in milk (DIM), milk yield 9.3 ± 1.4 kg) were used in 4 × 4 Latin square trial with 28 days experimental periods. Each experimental period consisted of 14 days of adaptation followed by 14 days of sample collection. Previous research has shown that supplemental bacteria and yeast do not colonize in the rumen, so no carry over from the previous period was expected and the 14 days adaption period would provide adequate time for the supplemental yeast to stabilize [28]. The health condition of the cows was monitored daily during the entire period. All cows were housed in individual indoor pens equipped with fans, individual feed bunk and water. Diets were fed as TMR with forage-to-concentrate ratio of 38:62 (dry matter basis, DM). Dietary ingredients and nutrients composition are shown in Table 1. The TMR diet contained AFB_1_ at a level of 22.28 µg/kg of TMR diet. This dose used for this trial was based on AFB_1_ concentrations that have been reported in diets fed to lactating dairy cows in our region [4] and were slightly above the 20 µg/kg limit established by U.S. FDA. A control diet without AFB_1_ was not included as prior research has demonstrated that AFM_1_ concentrations are positively correlated with dietary AFB_1_ concentrations and feeding less than 20 µg/kg does not result in AFM_1_ concentrations that exceed FDA limits [3]. Cows were individually fed at 05:00 and 15:00 h in amounts to allow ad libitum intake. The amount of TMR offered and refused was weighed daily for each cow to determine DMI. Cows also had unlimited access to water. Mineral and vitamin blocks were provided to cows for free-choice consumption. Treatments consisted of control (no supplemental yeast), *K. marxianus* CPY1 (K1Y), *K. marxianus* RSY5 (K2Y) or *P. kudriavzevii* YSY2 (PY). These yeasts were previously identified to have the potential to effectively detoxify aflatoxin in vitro [27]. Supplemental yeasts were individually fed at the rate of 2 g/day (1 g provided 1 × 10^9^ CFU) by mixing supplemental yeast with 20 g cornmeal based on our previous research [27]. Cows fed the control diet were fed 20 g corn meal without the yeast. To avoid contamination of the mixing equipment, the treatment mix was individually added on top of each cow’s TMR allotment during the morning feeding.

Samples of TMR and orts were collected on days 7, 11 and 14 of each period and pooled by treatment and stored at −20 °C. Composite samples were dried in the forced air oven at 60 °C for 3 days or until showing constant weight before grinding using a Wiley mill (Thomas Scientific, Swedesboro, NJ, USA) to sift through a sieve with 2 mm pore size. Samples dried at 100 °C for 24 h were analyzed for DM, crude protein (CP), ether extract (EE) and ash according to the Association of Official Analytical Chemists (AOAC) [29]. Concentrations of neutral detergent fiber (NDF) and acid detergent fiber (ADF) were determined according to Van Soest et al. [30]. Body weight was measured at the beginning of the trial and the end of each experimental period. Cows were milked twice daily at 05:00 and 15:00 h using a bucket milking machine (DeLaval, International AB, Tumba, Sweden). Milk yield was recorded for each cow at each milking using mechanical scales. Approximately 200 mL of milk was collected on days 12 and 13 of each period. One half of each sample was analyzed for total solid (TS), solid-not-fat (SNF), protein, fat and lactose using the MilkoScan 6000 instrument (Foss Electric, Hillerod, Denmark). The second half of the sample was stored at −20 °C for AFM_1_ analysis.

### 2.2. Analytical Procedures

AFB_1_ in feed was extracted by mixing 20 g of ground sample with 100 mL of extraction solvent (70% methyl alcohol) in an Erlenmeyer flask. The flask and its contents were shaken at 300 revolutions per min for 30 min and then allowed to stand for 5 min for setting of the slurry before harvesting the clear portion by filter paper (Whatman No.4). The filtrate was diluted (ratio 1:5) with 0.01 mol/L phosphate buffer to 1:20 (1 mL filtrate + 3 mL buffer) and analyzed using a DOA-Aflatoxin ELISA Test Kit (Sigma-Aldrich, St Louis, MO, USA) [31].

The AFM_1_ in milk was extracted using immunoaffinity columns in a complete high-performance liquid chromatography (HPLC) system (Class LC10, Shimadzu, Kyoto, Japan) consisting of an HPLC pump (LC-10AD), an auto injector (SIL-10A), a column oven (CTO-10A) and a fluorescence detector (RF-10AXL). A Spherisorb ODS-2 column (Waters Corporation, Milford, MA, USA) (5-µm inside diameter, 4 by 250 mm) with a C18 guard column (4 by 3 mm) and a column temperature of 40 °C for the mobile phase was utilized for the analysis. The manufacturer’s method was followed as previously described by Ruangwises and Ruangwises [32]. A 30 mL milk sample was transferred into a 30 mL plastic centrifuge tube and defatted by centrifugation at 3500 revolutions per min for 20 min at 25 °C. The resulting skimmed milk was placed in a 50 mL plastic syringe attached to the immunoaffinity column. The skimmed milk was allowed to flow onto the column at 1 mL/min by gravity. The column was washed twice with 3 mL of 0.01 M phosphate-buffered peptone and once with 20 mL of MilliQ water (Millipore Inc., Bedford, MA, USA) and then eluted with 2 mL of methanol and flushed with air. A 2 mL volume of eluate was filtered through a nylon filter (0.45 mL), evaporated to dryness with a nitrogen gas stream (50 °C) and dissolved with 400 µL of water–acetonitrilemethanol (40:35:25) to determine AFM_1_ concentrations. The calibration curve was prepared by plotting the peak area for each standard against the quantity of AFM_1_ injected. The equation of the calibration curve was used to compute the AFM_1_ content of the samples. The limit of detection (LOD) was determined based on the Q2B method of U.S. Food and Drug Administration [33]. The transfer of AFM_1_ in milk was calculated as the ratio between the AFM_1_ excreted in milk (µg/L) and the AFB_1_ intake (µg/kg) at the time when the toxin output in milk reached a steady state. This value was used to calculate the percentage of ingested AFB_1_ excreted daily in milk as AFM_1_:Transfer (%) = (Total AFM_1_ excreted (µg/day)/AFB_1_ ingested (µg/day)) × 100.

### 2.3. Statistical Procedures

The power analysis measure was conducted using PROC GLM power. Before data were subjected for statistical analysis, the variables were tested for normality using UNIVARIATE procedure. All data were analyzed as 4 × 4 Latin square using the analysis of variance (ANOVA) procedure in SAS version 9.4 software [34]. If significance was observed (when *p* < 0.05), treatment means were compared using the Duncan’s New Multiple Range Test (DMRT). The model used in this experiment is as follows:*Y*_ijk_ = *u* + *ρ*_i_ + *γ_j_ + τ_ij_* + *ε_ijk_*(1)
where:
*Y*_ijk_ = the measured variable;*u* = the overall mean;*ρ*_i_ = the effect of experimental period (i = 1, 2, 3, 4);*γ_j_* = the effect of animals (i = 1, 2, 3, 4);*τ_k_* = the effect of treatments (k = 1, 2, 3, 4);*ε_ijk_* = residue error.


## 3. Results and Discussion

### 3.1. AFM_1_ Concentration in Milk

AFB_1_ detoxification efficiencies of the yeast isolated from ruminal fluids were assessed in TMR diet containing AFB_1_ at a level of 22.28 µg/kg of TMR diet with the rate of 2 g/day (1 × 10^9^ CFU/g). Concentrations of AFM_1_ measured in milk from dairy cows fed diets contaminated with AFB_1_ are summarized in Table 2. All yeast-supplement treatments reduced (*p* < 0.05) the AFM_1_ transfer into milk compared with control, with K1Y and K2Y being most effective and PY intermediate. Concentrations of AFM_1_ in milk from the K1Y and K2Y were less than the maximum tolerance of 0.50 µg/kg established by the FDA [7], whereas AFM_1_ in control was 72-fold greater than the FDA [7] maximum tolerance level on the last day of the treatment period. The absorption of AFB_1_ from feed and conversion into AFM_1_ in milk was 1.54, 0.36, 0.43 and 0.51 µg/L in the control, K1Y, K2Y and PY group, respectively, and was reduced by the supplements (*p* < 0.01).

These results are consistent with previous reports in that AFB_1_ was readily absorbed within the gastrointestinal tract and metabolized in the liver to form AFM_1_, which is quickly excreted into milk or urine. As expected, cows fed AFB_1_ secreted substantial quantities of AFM_1_ into their milk [35,36,37]. Concentrations of AFM_1_ in milk were consistent with those reported by Kutz et al. [38], who also fed AFB_1_ at about 100 µg/kg of DM, but were markedly higher than those reported by others [13,28], who fed about 75 µg of AF/kg of DM, probably due in part to concentrations of AFB_1_ fed. The transfer rate observed in our trial is in agreement with those cited (1% to 6% in dairy cows) by the European Food Safety Authority [8]. The inverse relationship between transfer and AFB_1_ intake could be related to the biotransformation processes of these mycotoxins in animal tissues [1]. Kutz et al. [38] reported that supplementation of yeast cells at 0.50% of DM in feed containing 100.0 µg of AFB_1_/kg decreased AFM_1_ in milk from dairy cows. Battacone et al. [36] reported no effect of adding yeast cell wall and dried yeast supplements on AFM_1_ concentration in diets containing 60.0 µg of AFB_1_/kg. Ruminants have a complex metabolism and many factors that can influence the adsorption capacity of yeast in the gastrointestinal tract. One of the most important is the microbial composition of rumen fluid. However, these contradictory results regarding the transfer of AFB_1_ would be expected because gastrointestinal absorption and subsequent excretion as AFM_1_ in milk varies among animals because of nutritional and physiological factors, feed digestion, feeding regimens, animal health, hepatic biotransformation and milk yield [39].

### 3.2. Effect on Feed Intake, Nutrient Digestibility and Animal Performance

In the present study, 2.0 g/day (1 × 10^9^ CFU/g) of dry biomass of K1Y, K2Y and PY were supplemented to the TMR to examine the effect of AFB_1_-detoxifying yeast supplementation on production of lactating dairy cows. The DMI measured as kg/d, BW (%) and BW^0.75^ (g/kg) was greater (*p* < 0.05) in cows fed TMR supplemented with AFB_1_-detoxifying yeast compared with control (Table 3). The average DMI for cows fed supplemental yeasts was 13.77 kg/day compared with 9.54 kg/day for control. These results are consistent with previous research, in which supplemental yeast increased DMI [40,41,42]. Yeast cells are a rich source of vitamins, enzymes and other cofactors that stimulate microbial activity in the rumen, potentially increasing the amount of nutrients digested. Increasing digestion is a factor that contributes to improving DMI [43]. However, our results were not consistent with others who found no effect on DMI when cows received dietary AFB_1_ at 210 to 313 µg/kg and fed supplemental clay or inactivated yeast supplements to bind AFB_1_ [28,37]. Mycotoxin binders appear to have a dose-dependent effect on DMI [44].

Normally, apparent digestibility percentage declines slightly as DMI increases because of higher ruminal turnover; however, nutrient digestibility (Table 3) was not affected (*p* > 0.05) by aflatoxin-detoxifying yeast supplementation. The results of the present study are in agreement with Battacone et al. [36], who reported no differences in apparent total tract digestibility of DM, organic matter (OM), CP, EE, NDF, ADF and energy when dairy ewes were fed a dry yeast product (*Kluyveromyces lactis*). Jiang et al. [45] added *S. cerevisiae* fermentation product (SCFP) at 35 g/day of the dietary DM to diets containing 36.1% corn silage, 8.3% alfalfa hay and 55.6% concentrate (DM basis). These researchers reported that SCFP supplementation did not influence nutrient digestibility. The results of our present experiment indicated that supplementation of aflatoxin-detoxifying yeast at 2 g/day (1 × 10^9^ CFU/g) of DM did not affect total tract digestibility of DM, OM, CP, EE, NDF or ADF and are consistent with the previous reports.

No differences were observed in milk yield (*p* = 0.67) of cows fed aflatoxin-detoxifying yeast supplement (Table 4). However, supplementation of AFB_1_-detoxifying yeast enhanced ECM and yield and percentage of milk components (*p* < 0.01). Milk fat, protein, lactose, SNF and TS were highest in cows fed TMR supplemented with K1Y and K2Y, intermediate for PY and lowest for control, respectively (*p* < 0.01). Stroud et al. [46] observed that feeding diets containing 170.0 µg of AFB_1_/kg of DMI for more than 11 days had no effect on milk yield, even though AFB_1_ decreased feed intake 1.5 kg/day compared with cows fed diets without AFB_1_. Kutz et al. [38] and Mojtahedi et al. [47] reported that dairy cows fed 112.2 µg AFB_1_/kg of DM did not affect milk yield or composition. The supplementation of SCFP (*S. cerevisiae* fermentation product) consisting of yeast cells has been shown to increase DMI and milk yield, as well as milk fat and protein yield, in lactating dairy cows [42,45]. However, milk yield was decreased 3.0 L/day and concentration of milk fat and protein decreased after dairy cows were fed 43.5–120.0 µg of AFB_1_/kg of BW [48]. The reduction in milk protein concentration may be a consequence of the inhibition of microbial protein synthesis by AFB_1_ [28]. The result of this study validates the hypothesis that supplement yeast isolates improve aflatoxin detoxification and rumen fermentation, which would positively contribute to milk fat, protein and lactose syntheses. However, the limitation of this study was the small number of animals used (power of test = 0.69) to balance statistical power and potential concerns for animal welfare. Additional study is warranted to confirm the increase in feed intake, milk yield and milk components observed in the current study.

## 4. Conclusions

Supplementation of aflatoxin-detoxifying yeast (*K. marxianus* CPY1, RSY5 and *P. kudriavzevii* YSY2) demonstrated the capacity to detoxify aflatoxin B_1_ and decrease the transfer of aflatoxin B_1_ to aflatoxin M_1_ in raw milk by 72.08%. Moreover, our results indicated that *K. marxianus* CPY1 and RSY5 were most effective and have enormous potential for development into a supplement for mitigating aflatoxins commonly found in feed to improve food safety. The data also indicate that all yeast supplements fed may improve feed intake and milk component yield and percentages when fed at the rate of 2 g/day of DM.

## Figures and Tables

**Table 1 animals-10-00709-t001:** Ingredients and chemical composition of experimental diets on a DM basis.

Item	TMR Diet
Ingredient, % of DM	
Napier silage	38.00
Peanut meal	25.00
Coconut meal	15.50
Corn meal	21.00
Premix	0.50
Composition, % of DM ^1^	
TDN	73.39
DM	51.10
CP	16.95
EE	3.83
NDF	39.19
ADF	24.18
NFC	32.67
Ash	7.36
Aflatoxin B_1_, µg/kg	22.28

^1^ TDN: Total digestible nutrient calculated by ((digestible CP) + (digestible CF) + (digestible NFE) + (digestible EE x 2.25)); TMR: total mixed ration; DM: dry matter; CP: crude protein; EE: ether extract; NDF: neutral detergent fiber; ADF: acid detergent fiber; NFC: non-fiber carbohydrates calculated by difference (100 − (%NDF + %CP + %EE + Ash)).

**Table 2 animals-10-00709-t002:** Aflatoxin B_1_ (AFB_1_) dairy intake and least-squares means of concentration of aflatoxin M_1_ (AFM_1_) and carryover in cows fed TMR supplemented with aflatoxins-detoxifying yeast.

Item	Treatment ^1^				SEM	*p*-Value
	Control	K1Y	K2Y	PY		
AFB_1_ intake, µg/day	212.60	312.11	303.56	304.66		
AFM_1_, µg/L	1.54 ^a^	0.36 ^c^	0.43 ^c^	0.51 ^b^	0.12	<0.001
AFM_1_, µg/day	15.44 ^a^	3.68 ^b^	4.38 ^b^	5.15 ^b^	0.09	<0.001
Transfer ^2^, %	7.26 ^a^	1.18 ^b^	1.44 ^b^	1.69 ^b^	0.05	<0.001

^1^ Control: TMR with no supplemented yeast, K1Y: TMR supplemented with *K. marxianus* CPY1 10^9^ CFU/g, K2Y: TMR supplemented with *K. marxianus* RSY5 10^9^ CFU/g, and PY: TMR supplemented with *P. kudriavzevii* YSY2 10^9^ CFU/g; ^2^ transfer: percentage of AFB_1_ ingested (µg/day) that was converted to AFM_1_ and excreted in milk (µg/day); ^a,b,c^ means in the same row with different superscript were different at *p* < 0.05; SEM: standard error of the mean.

**Table 3 animals-10-00709-t003:** Dry matter intake (DMI), body weight (BW) change and nutrient digestibility of cows fed TMR supplemented with aflatoxin-detoxifying yeast.

Item	Treatment ^1^				SEM	*p*-Value
	Control	K1Y	K2Y	PY		
DMI, kg/day	9.54 ^b^	14.01 ^a^	13.62 ^a^	13.67 ^a^	0.29	0.01
DMI, %BW	2.46 ^b^	3.50 ^a^	3.53 ^a^	3.47 ^a^	0.18	0.05
DMI, g/kg BW ^0.75^	109.24 ^b^	156.63 ^a^	156.38 ^a^	154.48 ^a^	2.01	0.05
BW gain, kg/day	–0.23	0.05	0.07	0.03	2.36	0.44
Digestibility, %						
DM	61.37	63.28	63.55	60.92	1.05	0.44
OM	63.22	66.18	66.16	63.34	1.02	0.38
CP	69.68	71.11	72.03	68.97	1.41	0.25
EE	70.14	70.41	72.22	73.08	2.12	0.16
NDF	48.12	47.62	48.89	45.64	1.33	0.26
ADF	45.98	44.97	48.63	44.89	1.52	0.54
GE	61.58	63.26	62.59	61.07	2.15	0.15

^1^ Control: TMR with no supplemented yeast, K1Y: TMR supplemented with *K. marxianus* CPY1 10^9^ CFU/g; K2Y: TMR supplemented with *K. marxianus* RSY5 10^9^ CFU/g; and PY: TMR supplemented with *P. kudriavzevii* YSY2 10^9^ CFU/g; ^a,b,^ means in the same row with different superscript showed significant different at *p* < 0.05; SEM: standard error of the mean.

**Table 4 animals-10-00709-t004:** Milk yield and milk composition of cows fed TMR supplemented with aflatoxin-detoxifying yeast.

Item	Treatment ^1^				SEM	*p*-Value
	Control	K1Y	K2Y	PY		
Production, kg/day						
Milk yield	10.03	10.23	10.18	10.10	0.14	0.67
ECM ^2^	8.93 ^c^	11.27 ^a^	10.59 ^a,b^	9.84 ^a,b^	0.41	0.01
Fat	0.29 ^c^	0.43 ^a^	0.38 ^a,b^	0.35 ^b,c^	0.03	0.01
Protein	0.26 ^b^	0.33 ^a^	0.32 ^a^	0.28 ^b^	0.02	0.01
Lactose	0.43 ^b^	0.50 ^a^	0.50 ^a^	0.48 ^a^	0.01	0.01
Milk composition, %						
Fat	2.91 ^c^	4.19 ^a^	3.73 ^a,b^	3.47 ^b,c^	0.25	0.01
Protein	2.60 ^b^	3.22 ^a^	3.19 ^a^	2.76 ^b^	0.12	0.01
Lactose	4.33 ^c^	4.94 ^a^	4.96 ^a^	4.78 ^b^	0.11	<0.01
SNF ^3^	7.53 ^c^	8.95 ^a^	8.95 ^a^	8.16 ^b^	0.25	<0.01
Total solids	10.38 ^b^	13.08 ^a^	12.61^a^	11.59 ^a^	0.12	0.01

^1^ Control: TMR with no supplemented yeast, K1Y: TMR supplemented with *K. marxianus* CPY1 10^9^ CFU/g; K2Y: TMR supplemented with *K. marxianus* RSY5 10^9^ CFU/g; and PY: TMR supplemented with *P. kudriavzevii* YSY2 10^9^ CFU/g; ^2^ ECM: energy collected milk calculated from equation 0.327 × milk (kg) + 12.95 × fat (kg) + 7.20 × protein (kg); ^3^ SNF: solid not fat; ^a,b,c^ means in the same row with different superscript were different at *p* < 0.05; SEM: standard error of the mean.

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
