# Peer review of "Effect of Supplemental Kluyveromyces marxianus and Pichia kudriavzevii on Aflatoxin M1 Excretion in Milk of Lactating Dairy Cows"

_animals, 2020, doi:10.3390/ani10040709_

Round 1

Reviewer 1 Report

The manuscript entitled "Effect of Supplemental kluyveromyces marxianus and Pichia kudriavzevii isolated from ruminal fluid on aflatoxin M1 Excretion in Milk of Lactating Dairy Cow" is well written.

Few comments are reported here:

Summary, Abstract and Introduction

  • Line 20, Line 50, and Line 64: delete  "in Thailand" . The country of origin is not important in this part. This is a study presented to an International Journal.
  • Line 55-57; Line 64-66 and Line 77-80: Please, provide references at the end of the sentence.
  • Line 77-83 Add that Yeast help animals in performance and in rumenal ph: Armato et al., 2016 LARGE ANIMAL REVIEW; Armato et al., 2017 ACTA AGRICULTURAE SCANDINAVICA, SECTION A.

Materials and Method:

Comments:

- the title report "Effect of Supplemental Kluyveromyces marxianus and Pichia kudriavzevii Isolated from Ruminal Fluid...", please specify how this isolation from rumen liquid was set.

- it would be more appropriate to insert a table with .the ingredients of the diet and the chemical part of the diet. Once this is done, the excess of sentences in Experimental procedures  (2.1) could be eliminated.

- Authors need to improve or better specify the application of yeasts: mixer? top dress? it is not more clear in the text.

  • Line 97: are 14 days sufficient as a clerance period? Please insert a reference
  • Line 106: over the whole period? or in the periods included in the Latin square?
  • Line 147: SAS 9.4?

Reviewer 2 Report

Dear Authors,

I have reviewed the article titled “Effect of Supplemental Kluyveromyces marxianus and Pichia kudriavzevii Isolated from Ruminal Fluid on Aflatoxin M1 Excretion in Milk of Lactating Dairy Cows”. The objective was to assess the aflatoxin detoxifying potential of yeast isolated from lactating dairy cows included the diets on Aflatoxin B1, cow performance, and the content of aflatoxin m1 in raw milk.

I found this research novel and very interesting. This is a topic that needs to be researched because of its importance on food safety and animal production. In general terms, the research question is well aligned with the objectives and methods, and the manuscript is well written. However, I have significant concerns about the sample size, material and methods section, statistical analysis, and overambitious conclusions that I would like the authors to clarify and improve. I recommend giving the authors a chance to make the changes suggested by all reviewers and editors.

Please see my specific comments below:

L1-L5: With such a small sample size it is hard to determine a population effect of the supplemental yeast. However, this paper delivers important information for local and global dairy production. I would suggest changing the title and/or the scope of this research for a case report or analytical study.

L24: Where is detoxification measured? Blood? Liver? Milk? Please specify.

L25: Use DMI (kg/d) as shown in table 1 instead of feed intake %, 44.3% increase sounds shocking and I don’t think the sample size supports this conclusion because of within cow variation in DMI.

L34: add “subsequent” before supplementation.

L35: Why a negative control treatment was not used (no AFB1, no yeast)? I think this is a major flaw to extrapolate this conclusion to a larger population of dairy cows. This idea also supports my suggestion of considering this article as an analytical study or case report.

L58-59: delete “that have been”, add “with” before diets.

L65: add “,” after cow.

L71:  Is this the limit for the US, Thailand? Please clarify. Also, why did you choose a dose 11 times greater than the 20 ug/kg limit?

L88: Replace “was” by “were”.

L96: Please provide more information about the study population, farm estimate location, farm milk yield average, dates when the study was carried out, housing type, etc. This is especially needed.

L97: How long each supplemental period last? 14 days with 14 days for wash-off? Please clarify.

L99: A negative control (no AFB1, no yeast) would give a valuable comparison point. Please clarify, anywhere, why these 4 treatment groups were used.

L100: In the abstract, it was mentioned that diets contained 220 ug/d of AFB1, but this is not mentioned in the experimental procedures. Were all diets standardized to contain this amount every day? Did the diets contain added AFB1? I cannot figure this out with the information available in the manuscript. Please specify in the materials and methods section.

L100-101: Why the 2g/day dose, which provides 1*109 CFU/g was used. Was this driven from your previous study? Was it available in the literature? Please clarify.

L108: How feed intake was recorded? Did you weight the cornmeal mixture only, the full TMR? Were the cows individually fed? Did you use Calan gates? There are many details about practical procedures that need to be better explained.

L113: How the cows were milked? Were they the only ones being milked? I would like to know how the study cows were housed. Were they in a research pen just for them or were they comingled with more cows? Please be clear so I can have an idea of the research settings that provide information about why these results should be trusted or not. Was the milk yield automatically recorded?

L124: Please add the company maker of the ELISA kit.

L125: Please explain that the content of AFM1 was the outcome that showed detoxification. I could conclude this after reading the manuscript, but it should be explicitly stated.

L145: Statistical methods need to be improved. Did you check normality and other assumptions of ANOVA tests? With only 4 cows as experimental units, it seems that you would need to use a non-parametric test. Moreover, there are not standard errors in the results or in the tables. I am also concerned about not including DIM in the model. This is because you may have started with cows with 200 DIM and then finishing the study with over 300 DIM. Thus, the differences between treatments may be given by the normal lactation curve but not due to the treatment effect.

L145: How much power did you have with this sample size? Why did you use 4 cows? I can understand that costs and technical feasibility are limitations for the number of animals, but that should be explained to the reader and discussed as study limitations.

L163: What is BW change? I am not familiar with this method to measure DMI. Please clarify in materials and methods.

L166: This is DMI comparing control vs yeast (3 yeast group effect). I was shocked reading this considering that you only added 2g/d of a product. However, I think there are other factors not being controlled in this trial. For example, you could have had a negative control group across the experiment. Was the control group supplemented with the cornmeal? Please specify in the materials and methods.

L171: Did you cows received AFB1? I am confused at this point. Please clarify in the materials and methods section.

L173-179: This paragraph contradicts what is stated in L167-169.

L159: What was the p-value of the effect of experimental periods?

L210: This is the strongest paragraph of the manuscript. With the information presented you are able to prove that your yeast treatments work. I would change the scope of the paper by only focusing on these results, that in my eyes, are the most meaningful for the public health and the dairy industry. I don’t believe you have presented enough information to claim that supplemented yeast increased ECM, DMI, and milk components.

L244: I would reconsider the main conclusion of this manuscript. As I just said, the authors may have been overambitious claiming that yeast increased DMI and milk yield and solids, but I strongly believe that there is good information just restricted to the anti-aflatoxin effect of these yeasts on raw milk.

L249: What do you mean with “enormous”? Worldwide? Food safety or production-wise?

Round 2

Reviewer 2 Report

Dear Authors,

I truly believe the quality and clarity of the manuscript has greatly improved. However, I still have some points I would like the authors to go over and improve. Please see those comments below.

Point 1: I understand those reasons. Now that you have rearranged the structure of the manuscript to show detoxification results first, the title makes more sense.  Overall, the scope of the study is clearer now. However, I strongly recommend adding discussions about the validity of this study in terms of sample size, extrapolation to other population, not having a pure control diet, and confounder control.

L24-24: Please be consistent when presenting the results. Use the same order as you list the objectives in L22.

L30: Did you forget DMI as objective? Please include as you are presenting results of DMI in the abstract.

L35: Were all groups assigned to the AFB1 and the 2g of each (group 1 to 4) yeast biomass across the study? When you say 4 cows, it is not clear if this was performed across all groups or to an individual group. Please clarify accordingly.

L40: Please be consistent presenting the results in the manuscript. You can decide the order but keep it the same.

Point5: Please include that explanation and references in the introduction and/or methods or anywhere relevant.

L58: Use: “animals fed with diets containing” and delete “animals with before fed diets containing”.

L71: Here it might be a good place to add the references in point5.

Point8: Please specify this in the manuscript.

L98: Replace “Parity of 3” by “lactation number = 3”

Point10. Thanks for adding more details. Now the experimental settings are much clearer.

Point12. My question was not successfully addressed. I did not find this clarified in the text. Please add references that support your statements that normal diets have concentrations greater than 20ug/kg. Otherwise, to replicate this study in a commercial farm I would have to standardize the amount of AFB1 in the diets to expect similar results. What about diets containing way more or less?

Point14: Please add reference of add some data. I trust you but everything you state must be evidence based.

Point19: Add outlier assessment and power calculation to the statistical analyses. Did you test for normality? AFM1 may not by affected by DIM but I was thinking on the milk yield or milk component models. What was the P-value of the experimental period term in the milk yield model? In the tables, it is not clear to which group the SEM belongs to. Please explain in the table captions or add the SEM to each group and result. I recommend using a ± sign. E.g. mean ± SEM

L164: Please add all response variables analyzed by the model presented here.  

Point20: Please add this value into the discussion. I understand there is not perfect study and I believe this study provides valuable information. However, I think it is necessary to acknowledge and discuss the limitations and validity of this study.

Point25: Please add that information into the manuscript. I suggest to include it in section 3.2.Please use the exact P-value not P>0.05.

L244: Please add the exact value of the P-value.

L271: Replace “improved” by “may improve”.
